# Aurora Kinase A Regulates Cell Transitions in Glucocorticoid-Induced Bone Loss

**DOI:** 10.3390/cells12202434

**Published:** 2023-10-11

**Authors:** Xiaojing Qiao, Yang Yang, Yan Zhao, Xiuju Wu, Li Zhang, Xinjiang Cai, Jaden Ji, Kristina I. Boström, Yucheng Yao

**Affiliations:** 1Division of Cardiology, David Geffen School of Medicine at UCLA, Los Angeles, CA 90095-1679, USA; 2The Molecular Biology Institute at UCLA, Los Angeles, CA 90095-1570, USA

**Keywords:** glucocorticoid, bone loss, osteoblast, transition, endothelial-like cells, aurora kinase

## Abstract

Glucocorticoid-induced bone loss is a severe and toxic effect of long-term therapy with glucocorticoids, which are currently prescribed for millions of people worldwide. Previous studies have uncovered that glucocorticoids reciprocally converted osteoblast lineage cells into endothelial-like cells to cause bone loss and showed that the modulations of Foxc2 and Osterix were the causative factors that drove this harmful transition of osteoblast lineage cells. Here, we find that the inhibition of aurora kinase A halts this transition and prevents glucocorticoid-induced bone loss. We find that aurora A interacts with the glucocorticoid receptor and show that this interaction is required for glucocorticoids to modulate Foxc2 and Osterix. Together, we identify a new potential approach to counteracting unwanted transitions of osteoblast lineage cells in glucocorticoid treatment and may provide a novel strategy for ameliorating glucocorticoid-induced bone loss.

## 1. Introduction

Millions of adults worldwide are prescribed glucocorticoids for long-term use [1,2]. Osteoporosis followed by severe fractures is one of the most prevalent side effects of this long-term therapy [1,2,3]. Interestingly, the risk of fracture is dramatically increased after initial the glucocorticoid treatment, worsens as treatment continues, and peaks after one year of therapy [2]. Higher doses of glucocorticoids result in more severe osteoporosis and fractures [2,3].

Osteoblasts that reside in the endosteum and periosteum have some plasticity for bone formation [4,5,6] and can be driven into non-osteogenic lineages [7,8,9,10,11,12,13,14]. Osteoblasts and bone endothelial cells closely adjoin and regulate each other to couple osteogenesis and angiogenesis [15,16,17,18,19]. Endothelial cells are known to transform into osteoblast-like cells in several medical conditions [20,21,22]. In a previous study, it was found that glucocorticoid converted osteoblast lineage cells into endothelial-like cells to cause osteoporosis [23]. Using a glucocorticoid-induced mouse model and osteoblast lineage tracing, the study showed that endothelial markers emerge in the osteoblast lineage cells of glucocorticoid-induced bone loss. After the cell transplantation experiments, the study uncovered that glucocorticoid-treated osteoblasts lose their osteogenic capacity in ectopic bone formation but improve their vascular repair capacity. Mechanistically, the study showed that the coupled alterations in Foxc2 and Osterix are responsible for the shift of osteoblast lineage cells into endothelial differentiation [23].

In this study, we identify that the inhibition of aurora A significantly improves glucocorticoid-induced bone loss. We uncover that glucocorticoid increases the interaction of aurora A with the glucocorticoid receptor, and this interaction is required for glucocorticoid to induce or repress Foxc2 and Osterix levels, respectively. Aurora A inhibition interrupts this physical interaction and prevents osteoblast lineage cells from performing endothelial differentiation.

## 2. Methods

### 2.1. Animals

*Wild-type* (C57BL/6J), *Col1α1^CreERT2^* (B6.Cg-Tg(Col1α1-cre/ERT2)1Crm/J), *Rosa^tdTomato^* (B6;129S6-Gt(ROSA)26Sortm9(CAG-tdTomato)Hze/J), and *AuroraA^flox/+^* (B6.129-Aurkatm1.1Tvd/J) mice on C57BL/6J background were obtained from the Jackson Laboratory. PCR was performed to confirm genotypes [24]. F4–F6 generations were utilized for experiments. We used littermates as controls. Since glucocorticoid-induced osteoporosis increases the incidence of fractures in both men and women [1,2], mixed genders of mice were used in each experimental group. We fed the mice with a standard chow diet (Diet 8604, HarlanTeklad Laboratory, Indianapolis, IN, USA). Mice received dexamethasone (0.08 µg/g, daily) at 8 weeks of age for 4 weeks as previously described [25,26]. The Institutional Review Board reviewed and approved the studies, which were conducted in accordance with the animal care guidelines set by the University of California, Los Angeles (UCLA). The studies conformed to the National Research Council, Guide for the Care and Use of Laboratory Animals, Eighth Edition (Washington, DC, USA: The National Academies Press, 2011).

### 2.2. Tissue Culture

The osteoblast cell lines MC3T3 and hFOB1.19 were purchased from American Type Culture Collection (ATCC, CRL-2593, and CRL11372) and cultured as per the manufacturer’s protocol. Dexamethasone (Sigma-Aldrich, D1756, St. Louis, MO, USA) and Betamethasone (Sigma-Aldrich, B7005, St. Louis, MO, USA) treatments were performed as described in the experiments. Lentiviral vectors containing CMV-aurora A and CMV-glucocorticoid receptor were all purchased from GeneCopeia^TM^ (Rockville, MD, USA) and applied to the cells as per the manufacturer’s protocols.

### 2.3. Micro-CT Imaging

Femurs were isolated at the acetabulum of mice and cleaned from the tibias. Then, the muscle was cleaned. The cleaned femurs were fixed in 4% (wt/vol) paraformaldehyde (PFA) for 24 h. Then, 50% alcohol was used to replace PFA, and the femurs were ready for Micro-CT imaging, which was performed at the Crump Imaging Center at UCLA as described [23].

### 2.4. Histomorphometric Analysis

Mean intercept length method: With the original mean intercept length (MIL) method, test lines and their intercepts are traced and counted. Mean intercept length was divided by the test-line length. Surface-to-volume ratio is determined via MIL. A model of bone and trabecular number thickness was set, and separation was derived. The principal directions of ellipsoid and the degree of anisotropy were analyzed by the distribution of the MILs. Connectivity Density: The Conn–Euler method of Odgaard was used to examine connectivity density. The surfaces of cubic AIMs were examined via boundary/edge problem suppression. We used GOBJ to calculate the connectivity of the complete bone or to put an artificial cortex around the spongiosa to measure intact bones. Structure Model Index: The infinitesimal amount was used to dilate the triangulated surface. The new bone surface and volume were calculated to determine the derivative of the bone surface. The SMI was analyzed for a model type, such as flat plates having an SMI of 0, cylindrical rods having an SMI of 3, and round spheres having an SMI of 4. Values below 0 were considered ’air bubbles’ inside the bone. Distance Transformation Methods: The value of the radius of the maximal sphere was examined to calculate trabecular thickness with twice the mean value of all structure. Voxels was referred to as the mean thickness. The bone and background were switched with the same procedure to calculate trabecular separation. The spacing of the mid-axes for trabecular separation were examined to generate the inverse of the mean spacing to calculate trabecular number. Triangularization of surface method: This method directly examined the surface-to-volume ratio determined via the MIL method to determine the number, the thickness, and the separation of the trabecular. The area of the surface triangles was projected onto a directional surface distribution unaffected by changes of area-weighted normal vector with the di-rection distribution to calculate the principal directions of the MIL ellipsoid and the degree of anisotropy.

### 2.5. RNA Analysis

Real-time PCR analysis was performed as previously described [24,27]. Glyceraldehyde 3-phosphate dehydrogenase (GAPDH) was utilized in experiments as a control gene [24]. We purchased the primers and probes for mouse Osterix, Osteocalcin, Osteopontin, Foxc2, Flk1, VE-cadherin, and CD31 from Applied Biosystems (Waltham, MA, USA) as part of Taqman^®^ Gene Expression Assays.

### 2.6. Fluorescence-Activated Cell Sorting (FACS)

FACS analysis was performed as previously described [24]. A syringe was used to flush out bone marrow from femurs with phosphate-buffered saline (PBS). After that, femurs were cut into small pieces, which were incubated in collagenase solution (collagenase I and collagenase II mixture) for 25 min at 37 °C. After the incubation, the digested mix was collected and washed with cold PBS three times. The collected cells were filtered through cell strainers (70 µm and 40 µm) and centrifuged at 300× *g*. A total of 2 mL red blood cell lysis buffer was added to the precipitated cells for 10 min. We collected the cells again and washed them with PBS three times. After this, the cells were ready for staining. The cells were stained with fluorescein isothiocyanate (FITC)-, phycoerythrin (PE)-, or Alexa Fluor 488 (AF-488)-conjugated antibodies against CD31 and VE-cadherin (all from BD biosciences, 550274 and 562243, Franklin Lakes, NJ, USA) and Osteocalcin (ThermoFisher Scientific, 23418, Waltham, MA, USA). Nonspecific fluorochrome- and isotype-matched IgGs (BD Pharmingen, Franklin Lakes, NJ, USA) served as controls. A gating strategy was utilized as we used before [24]. Forward scatter (FSC) and side scatter (SSC) were used to set up the gates and regions, and cell populations were analyzed using specific markers.

### 2.7. Immunoblotting and Immunofluorescence

Immunoblotting was performed as previously described [20]. Equal amounts of tissue lysates were used for immunoblotting. Blots were incubated with specific antibodies to Foxc2, CD31, and aurora (Abcam, ab5060, ab182982 and ab13824, Cambridge, UK), Osterix (Santa Cruz Biotechnology (Dallas, TX, USA), sc-22536), Osteocalcin, and glucocorticoid receptor (ThermoFisher Scientific, PA1-511A and MA1-510). β-actin (Sigma-Aldrich, A2228) was used as a loading control.

### 2.8. Immunofluorescence

Immunofluorescence was performed as previously described [28]. Cleaned bone tissues were fixed with 4% (wt/vol) ice-cold paraformaldehyde (PFA) solution for 4 h. Then, the fixed bones were washed with PBS three times to remove PFA. After that, bones were incubated in cold 0.5 M EDTA, pH 7.4–7.6, for 24 h for decalcification. The bones were washed with PBS, and then incubated in cryoprotectant (CPT) solution at 4 °C for 24 h. Then, the decalcified bones were embedded and cryosectioned for staining. We used specific antibodies to CD31 (BD Bioscience, 553370, Franklin Lakes, NJ, USA) and Osterix (Santa Cruz Biotechnology, sc-22536). 4′,6-diamidino-2-phenylindole (DAPI, Sigma-Aldrich, D9564) was used for staining of nuclei.

### 2.9. ChIP-Assay

ChIP assays were performed as previously described [29]. We used specific antibodies for aurora A (Abcam, ab1287, Cambridge, UK). To target the DNA-binding sites of glucocorticoid receptor, the following primers were used for real-time PCR: 5′-tagggacatctcggtgcttc-3′ and 5′-ggagttccccgaccttcaag-3′ in the Foxc2 promoter; 5′-cacactgccaccctgaacta-3′ and 5′-aacacggagatgatgcatgc-3′ in the Osterix promoter.

### 2.10. Statistical Analysis

GraphPad Instat^®^, version 3.0 (GraphPad Software), was used for statistical analysis. Either unpaired 2-tailed Student’s *t* test or one-way ANOVA with Tukey’s multiple-comparisons test was used to analyze for statistical significance. We used pwr R package to calculate the size of the effect and the statistical power as previously described [23]. The number of animals in each group was sufficient to reach more than 80% power to identify differences in the results. The n represented the number of animals that were used in each experimental group.

## 3. Results

### 3.1. The Inhibitor of Aurora A Prevents a Glucocorticoid-Induced Osteoblast-Endothelial Transition

In a compound screen study, we found that the inhibitor of aurora kinase VX680 is a potential candidate for preventing a glucocorticoid-induced osteoblast-endothelial transition. When we treated mouse osteoblasts (MC3T3-E1) with dexamethasone in combination with the aurora kinase inhibitor VX680, we found that VX680 dose-dependently inhibited the induction of endothelial markers and re-instated the expression of osteogenic markers (Figure 1a). To determine if VX680 altered the effect of dexamethasone in human osteoblasts, we used a human osteoblast line (hFOB1.19) and treated the cells with dexamethasone and VX680. Real-time PCR showed that VX680 prevented the induction of endothelial markers and the reduction of osteogenic markers (Figure 1b), suggesting that VX680 also inhibited the effect of dexamethasone in human osteoblasts. VX680 inhibits the activity of aurora kinase A, B, and C [30]. To determine which aurora is involved in glucocorticoid-induced shifts of osteoblasts, we used specific siRNA to individually deplete aurora A, B, and C in dexamethasone-treated osteoblasts. Only the knockdown of aurora A inhibited the alterations in Foxc2 and Osterix (Figure 1c), supporting an important role of aurora A in glucocorticoid-induced shifts of osteoblasts toward endothelial differentiation.

### 3.2. Aurora A Is Involved in the Modulations of Foxc2 and Osterix by Glucocorticoid through Interaction with the Glucocorticoid Receptor

We treated the osteoblasts with dexamethasone in combination with VX680 or transfection of aurora A siRNA. After 24 h, we collected the cells and performed co-immunoprecipitation followed by immunoblotting using specific antibodies to aurora A or the glucocorticoid receptor. The results showed that a precipitated complex interacted with both anti-aurora A and antiglucocorticoid receptor antibodies. VX680 or aurora A siRNA interrupted these interactions (Figure 2a), suggesting that aurora A bound directly to the glucocorticoid receptor, and VX680 or aurora A siRNA abolished this binding. We located a glucocorticoid receptor DNA-binding site at −1687 bp upstream of the *Foxc2* gene and −1050 bp upstream of the *Osterix* gene. We performed a ChIP assay using dexamethasone-treated osteoblasts and showed an enrichment of aurora A at DNA-binding sites of the glucocorticoid receptor in the promoters of *Foxc2* and *Osterix* (Figure 2b). Again, VX680 abolished the enrichment. The results suggested that the interaction between aurora A and the glucocorticoid receptor directly targeted *Foxc2* and *Osterix* for transcriptional regulation.

We directly increased protein levels of aurora A and the glucocorticoid receptor in osteoblasts using lentiviral vectors containing CMV-aurora A or a CMV-glucocorticoid receptor. We found that individually increased aurora A or glucocorticoid receptor did not change the expression of Foxc2 and Osterix. Only the elevation of both modulated the expression of Foxc2 and Osterix (Figure 3a). Moreover, in the presence of dexamethasone, excess glucocorticoid receptor induced aurora A to enhance Foxc2 and abolish Osterix. Overexpression of aurora A also enhanced the effects of dexamethasone on the modulation of Foxc2 and Osterix. The combination of excess aurora A and glucocorticoid receptor escalated dexamethasone activity to the highest level, thereby furthering the Foxc2 induction and abolishing Osterix (Figure 3b). Together, the results suggested that the interaction between aurora A and the glucocorticoid receptor is required for glucocorticoids to modulate Foxc2 and Osterix and shift osteoblasts to endothelial differentiation (Figure 3b).

### 3.3. Aurora A Inhibition Ameliorates Glucocorticoid-Induced Bone Loss

To determine whether aurora A inhibition affected glucocorticoid-induced osteoporosis, we treated mice with dexamethasone (0.08 µg/g, daily) in combination with VX680 (5 µg/g, daily) for 4 weeks. Micro-CT showed that the VX680 treatment significantly improved the bone volume fraction, trabecular thickness, trabecular number, and trabecular spacing of the femurs of dexamethasone-treated mice (Figure 4a,b). The results suggested that aurora A inhibition reduced glucocorticoid-induced bone loss.

We examined the femurs using immunostaining. We found a reduction in the endothelial marker CD31 with an induction of Osterix and normalized trabecular structure in VX680-treated mice (Figure 5a). We isolated the cells of demarrowed femurs and examined them using FACS, which showed a decreased number of cells that co-expressed endothelial and osteoblastic markers in the VX680-treated mice (Figure 5b). We collected the demarrowed femurs and examined the gene expression using real-time PCR, which showed a normalization of endothelial and osteoblastic markers in the VX680-treated group (Figure 5c).

### 3.4. Aurora A Deletion Ameliorates Glucocorticoid-Induced Bone Loss

We created *Col1a1^cre/ERT2^AuroraA^flox/flox^* mice. At 6 weeks of age, we treated the *Col1a1^cre/ERT2^AuroraA^flox/flox^* mice with tamoxifen (75 µg/g, daily) for 5 consecutive days to reduce the expression of aurora A in osteoblast lineage cells. Non-tamoxifen treatment was used as a control. At 8 weeks of age, the mice received subcutaneous injection of dexamethasone (0.08 µg/g, daily) for 4 weeks. The results of micro-CT showed that tamoxifen administration to the *Col1a1^cre/ERT2^AuroraA^flox/flox^* mice significantly prevented dexamethasone-induced bone loss (Figure 6a). In the non-dexamethasone treated groups, no effect of tamoxifen administration was found in the *Col1a1^cre/ERT2^AuroraA^flox/flox^* mice compared with the controls (Figure 6a). We further performed histomorphometric analysis of these mice. First, total volume, bone volume, and relative bone volume (bone/total bone volume) were examined via the mean intercept length method [31]. Connectivity density was examined by the Conn–Euler method of Odgaard [32]. The trabecular numbers, thickness, and separation were examined via distance transformation methods [33]. Then, we validated the results by triangulating the surface of the segmented object and calculating the volume of the enclosed tetrahedrons and the surface of the triangles (a detailed description can be found in the supplemental information) [33]. The data showed the improvements in total volume, bone volume, relative bone volume, connectivity density, bone surface, trabecular numbers, thickness, and separation in the *Col1a1^cre/ERT2^AuroraA^flox/flox^* mice with tamoxifen administration compared with non-tamoxifen-treated controls after dexamethasone treatment (Figure 6b).

We isolated the cells of demarrowed femurs of these mice. FACS showed a decreased number of cells that co-expressed VE-cadherin and Osteocalcin in the *Col1a1^cre/ERT2^AuroraA^flox/flox^* mice with tamoxifen administration compared with the non-tamoxifen-treated controls after dexamethasone treatment (Figure 7a). We then isolated the Osteocalcin-positive cells, lysed the cells, and performed immunoblotting. The results showed an efficient deletion of aurora A in the *Col1a1^cre/ERT2^AuroraA^flox/flox^* mice with tamoxifen administration (Figure 7b). The deletion of aurora A prevented the induction of Foxc2 and CD31 and inhibited the decrease in Osterix and Osteopontin in the *Col1a1^cre/ERT2^AuroraA^flox/flox^* mice with dexamethasone treatment (Figure 7b). Using Osteocalcin-positive cells, we performed ChIP assays and showed that the deletion of aurora A depleted an enrichment of aurora A at DNA-binding sites of the glucocorticoid receptor in the promoters of *Foxc2* (Figure 7c) and *Osterix* (Figure 7d) of dexamethasone-treated *Col1a1^cre/ERT2^AuroraA^flox/flox^* mice. Together, the results suggested that aurora A inhibition prevented glucocorticoids from driving osteoblast lineage cells to endothelial differentiation and reduced osteoporosis.

## 4. Discussion

In a recent study, it was shown that a shift of osteoblast lineage cells toward endothelial differentiation was induced by glucocorticoids and that this unwanted cell transition caused osteoblast lineage cells to lose osteogenic capacity, thereby contributing to glucocorticoid-induced bone loss [23]. The study also showed that glucocorticoids suppress Osterix expression but activate Foxc2 in osteoblast lineage cells to switch the cell fate and functional capacity, contributing to osteoporosis [23]. Here, we unveil a new interaction between the glucocorticoid receptor and aurora A in controlling the plasticity of osteoblasts. As our data suggest, the direct binding of the glucocorticoid receptor and aurora A modulates Foxc2 and Osterix to cause an osteoblastic–endothelial transition. The results suggest the essential role of aurora A in glucocorticoid-induced osteoporosis and highlight the importance of aurora A inhibition in reducing the toxic effects of glucocorticoids on bone.

Aurora kinases are identified as phosphotransferases and essential for cell proliferation and differentiation [34]. Aurora kinases include three family members, aurora A, B, and C, which control chromatid segregation and genetic stability [35]. Aurora kinases regulate the activity of transcription factors. Aurora A phosphorylates Twist1, Yes-associated protein, and Yin Yang 1 to modulate their transcriptional activity [36,37,38]. Aurora A also binds directly to the transcription factors Foxo3a and Foxm1 in order to balance their activity [39,40]. VX680 specifically binds to aurora A via π-π interactions to inhibit its activity [30]. In this study, our results suggest that aurora A directly binds to the glucocorticoid receptor to induce Foxc2 for endothelial differentiation and decrease Osterix for suppression of osteoblastic differentiation. This investigation unveils a new interaction between the glucocorticoid receptor and aurora A and identifies aurora A as a key component that balances the activity of glucocorticoids in osteoblastic and endothelial differentiation. The inhibition of aurora kinases may lead to new therapeutic strategies that target osteoblastic differentiation and reduce the toxic effects of glucocorticoids on bone.

## Figures and Tables

**Figure 1 cells-12-02434-f001:**
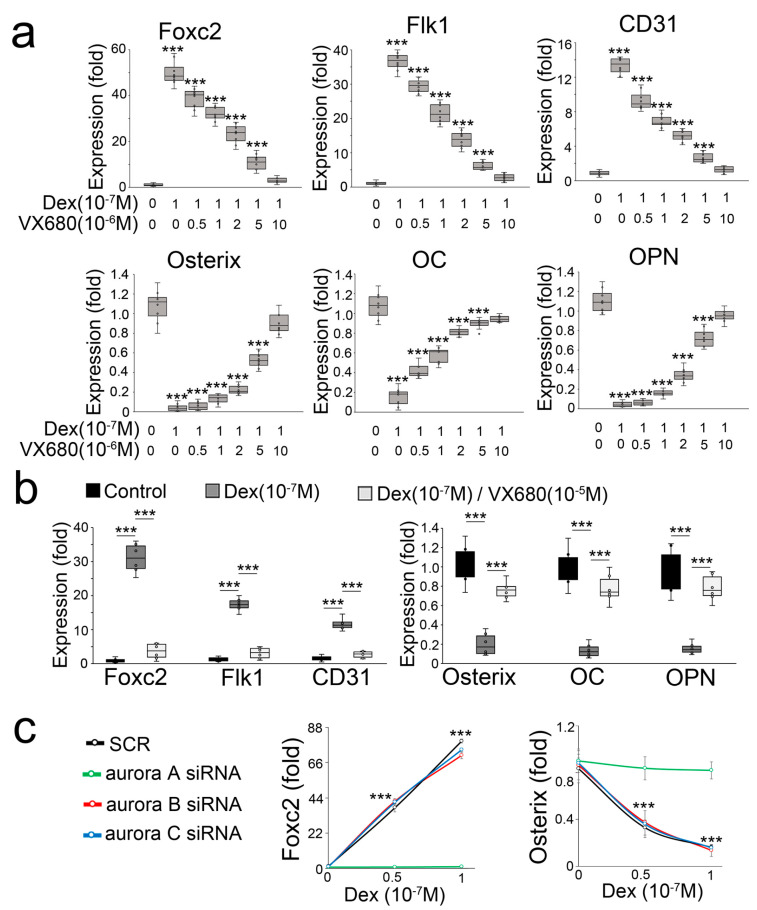
Aurora A inhibition prevents glucocorticoid from shifting osteoblasts to endothelial differentiation. (**a**) Expression of endothelial and osteogenic markers in mouse osteoblasts treated with dexamethasone (Dex) in combination with aurora kinase inhibitor VX680 (n = 8); (**b**) Expression of endothelial and osteogenic markers in human osteoblasts treated with Dex in combination with VX680 (n = 6); (**c**) Expression of Foxc2 and Osterix in dexamethasone-treated osteoblasts after knockdown of aurora A, B, or C (n = 5). Data were analyzed for statistical significance through ANOVA with post hoc Tukey’s analysis. The bounds of the boxes are upper and lower quartiles with data points. The line in the box is median. Error bars are maximal and minimal values. OC, Osteocalcin. OPN, Osteopontin. ***, *p* < 0.0001.

**Figure 2 cells-12-02434-f002:**
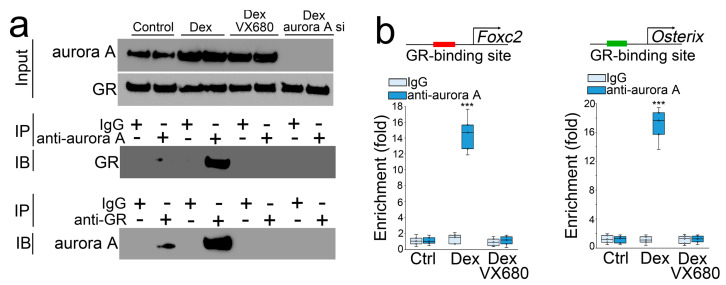
Aurora A is involved in the modulation of Foxc2 and Osterix by glucocorticoid through interaction with the glucocorticoid receptor. (**a**) Co-immunoprecipitation followed by immunoblotting using the antiglucocorticoid receptor or anti-aurora A antibodies. IP—immunoprecipitation. IB—immunoblotting; (**b**) ChIP assay using anti-aurora A antibodies in osteoblasts treated by dexamethasone with or without VX680 (n = 5). (**b**) was analyzed for statistical significance through ANOVA with post hoc Tukey’s analysis. The bounds of the boxes are upper and lower quartiles with data points. The line in the box is median. ***, *p* < 0.0001.

**Figure 3 cells-12-02434-f003:**
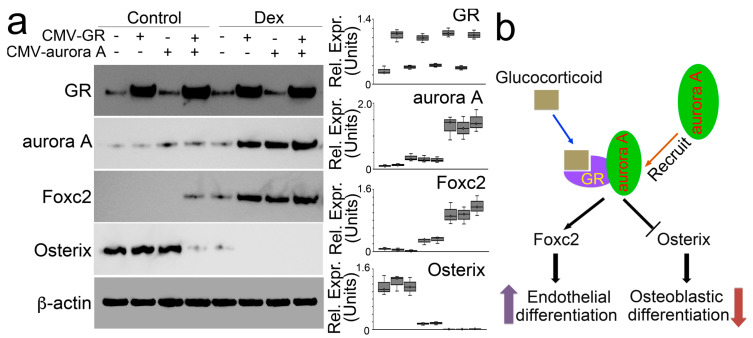
Aurora A is involved in the modulation of Foxc2 and Osterix by glucocorticoid through interaction with the glucocorticoid receptor. (**a**) Immunoblotting with densitometry analysis of glucocorticoid receptor (GR), aurora A, Foxc2, and Osterix in dexamethasone-treated osteoblasts after overexpression of glucocorticoid receptor (CMV-GR) or aurora A (CMV-aurora A). b-actin was used as loading control; (**b**) A schematic working model to show how glucocorticoid and glucocorticoid receptor recruit aurora A to regulate the expression of Foxc2 and Osterix.

**Figure 4 cells-12-02434-f004:**
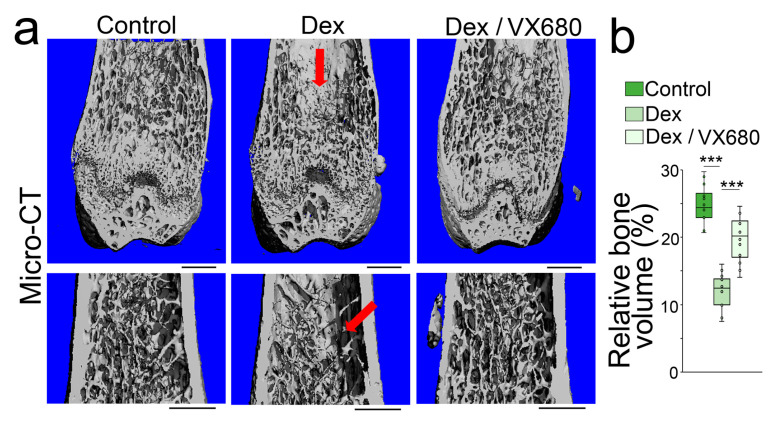
Aurora A inhibition ameliorates glucocorticoid-induced bone loss. *(***a**) Micro-CT (top) and immunostaining (bottom) of the femurs of the mice treated with dexamethasone with or without VX680 (top) (n = 12). Scale bar, 0.5 mm for micro-CT, 50 µm for immunostaining; (**b**) Relative bone volume of mouse femurs after dexamethasone treatment with or without VX680 (n = 12). (**b**) was analyzed for statistical significance using ANOVA with post hoc Tukey’s analysis. The bounds of the boxes are upper and lower quartiles with data points. The line in the box is the median. ***, *p* < 0.0001.

**Figure 5 cells-12-02434-f005:**
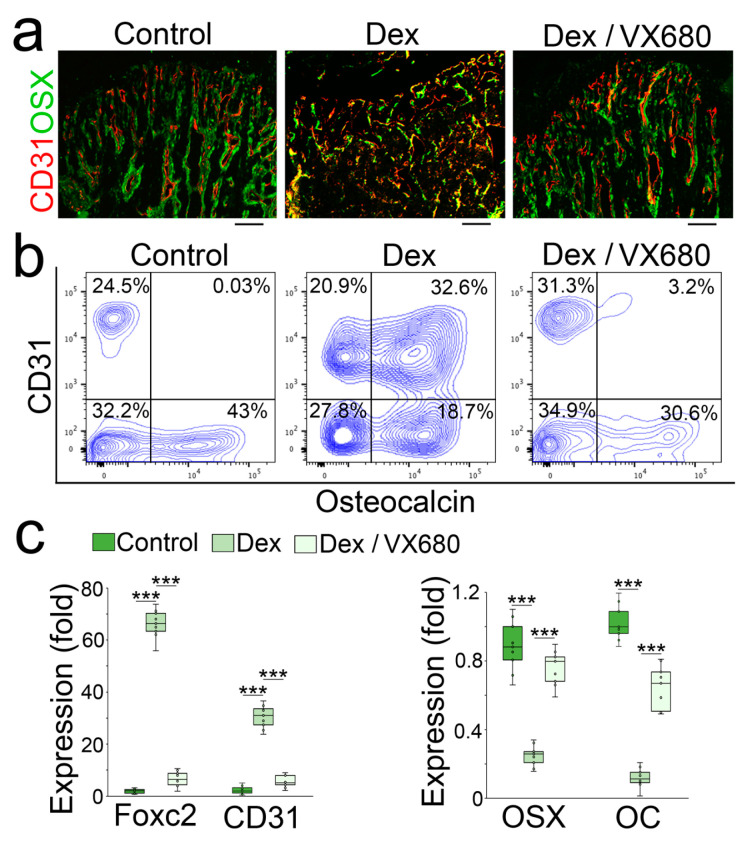
Aurora A inhibition prevents glucocorticoid from shifting osteoblasts to endothelial differentiation in glucocorticoid-treated mice. (**a**,**b**) Immunostaining (**a**) and FACS analysis (**b**) of demarrowed femurs of dexamethasone-treated mice with or without VX680 using anti-CD31 and anti-Osteocalcin antibodies; (**c**) Expression of endothelial and osteogenic markers in demarrowed femurs of dexamethasone-treated mice with or without VX680 (n = 9). (**c**) was analyzed for statistical significance using ANOVA with post hoc Tukey’s analysis. The bounds of the boxes are upper and lower quartiles with data points. The line in the box is the median. ***, *p* < 0.0001.

**Figure 6 cells-12-02434-f006:**
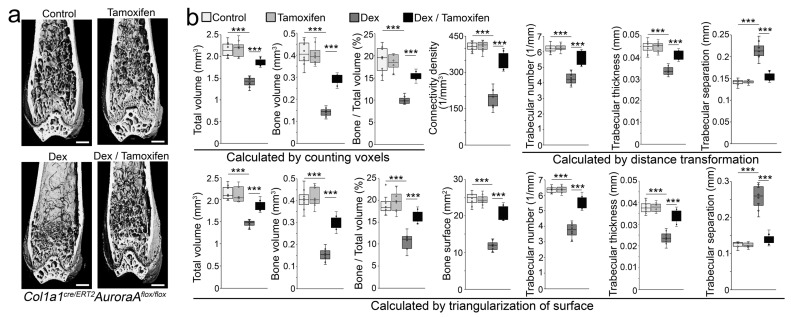
Inducible deletion of aurora A in osteoblasts inhibits glucocorticoid-induced bone loss. (**a**) Micro-CT of the femurs of *Col1a1^cre/ERT2^AuroraA^flox/flox^* mice after tamoxifen and dexamethasone treatment (n = 8). Scale bar, 0.5 mm for micro-CT. ***, *p* < 0.0001; (**b**) Histomorphometric analysis of the femurs of *Col1a1^cre/ERT2^AuroraA^flox/flox^* mice after tamoxifen and dexamethasone treatment (n = 8). Scale bar, 0.5 mm for micro-CT. (**b**) was analyzed for statistical significance using ANOVA with post hoc Tukey’s analysis. Error bars are maximal and minimal values. ***, *p* < 0.0001.

**Figure 7 cells-12-02434-f007:**
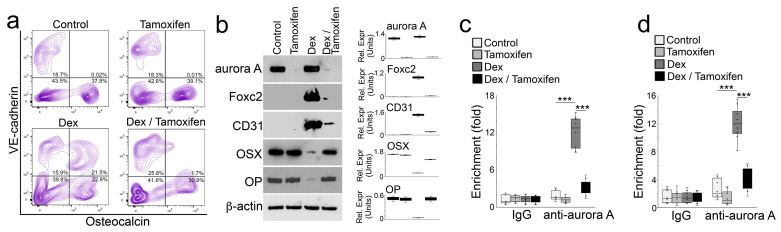
Inducible deletion of aurora A in osteoblasts inhibits glucocorticoid-induced bone loss. (**a**) FACS analysis of demarrowed femurs of *Col1a1^cre/ERT2^AuroraA^flox/flox^* mice with tamoxifen and dexamethasone treatment using anti-VE-cadherin and anti-Osteocalcin antibodies; (**b**) Immunoblotting with densitometry analysis of aurora A, Foxc2, CD31, Osterix (OSX), and Osteopontin in Osteocalcin-positive cells isolated from demarrowed femurs of *Col1a1^cre/ERT2^AuroraA^flox/flox^* mice with tamoxifen and dexamethasone treatment. β-actin was used as loading control; (**c**,**d**) ChIP assay at glucocorticoid receptor DNA-binding site of the *Foxc2* gene (**c**) and the Osterix gene (**d**) using anti-aurora A antibodies in Osteocalcin-positive cells isolated from demarrowed femurs of *Col1a1^cre/ERT2^AuroraA^flox/flox^* mice with tamoxifen and dexamethasone treatment (n = 8). (**c**,**d**) were analyzed for statistical significance using ANOVA with post hoc Tukey’s analysis. Error bars are maximal and minimal values. ***, *p* < 0.0001.

## Data Availability

Not Applicable.

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
