# Peer review of "Aurora Kinase A Regulates Cell Transitions in Glucocorticoid-Induced Bone Loss"

_cells, 2023, doi:10.3390/cells12202434_

Round 1
Reviewer 1 Report
Qiao et. al. provide results that are consistent with Aurora kinase A-mediated regulation of the transition by osteoblasts to endothelial-like cells. Using both in vitro and in vivo strategies, findings were secured that inhibition of aurora kinase A results in a decrease of glucocorticoid-induced bone loss. Mechanistically, the findings advance understanding of osteoblast plasticity and from a clinical perspective, there is potential for treatment of patients who undergo long-term and at times high dose, glucocorticoid treatment. While the data is consistent with the conclusions of the authors, the biological and clinical potential should be reinforced by expanding the gene expression analysis to an RNA-seq approach and validation of the mouse studies by human osteoblast analysis.
This paper expands understanding for glucocorticoid-compromised skeletal homeostasis and results are discussed within the context of an extensive series of published findings. There's appropriate consideration for data presented in relation to previous work and appropriate citations are provided.

Author Response
Reviewer 1
“Qiao et. al. provide results that are consistent with Aurora kinase A-mediated regulation of the transition by osteoblasts to endothelial-like cells. Using both in vitro and in vivo strategies, findings were secured that inhibition of aurora kinase A results in a decrease of glucocorticoid-induced bone loss. Mechanistically, the findings advance understanding of osteoblast plasticity and from a clinical perspective, there is potential for treatment of patients who undergo long-term and at times high dose, glucocorticoid treatment. While the data is consistent with the conclusions of the authors, the biological and clinical potential should be reinforced by expanding the gene expression analysis to an RNA-seq approach and validation of the mouse studies by human osteoblast analysis.
This paper expands understanding for glucocorticoid-compromised skeletal homeostasis and results are discussed within the context of an extensive series of published findings. There's appropriate consideration for data presented in relation to previous work and appropriate citations are provided.”
Thank you for the comments. To determine if VX680 altered the effect of dexamethasone in human osteoblasts, we used a human osteoblast line (hFOB1.19) and treated the cells with dexamethasone and VX680. Real-time PCR showed that VX680 prevented the induction of endothelial markers and the reduction of osteogenic markers (Figure 1b), suggesting that VX680 also inhibited the effect of dexamethasone in human osteoblasts. The results were added to Figure 1b and the main text.
This study focuses on the specific effect of aurora A on the transition of endothelial cells in glucocorticoid-induced bone loss. We have demonstrated interaction of aurora A with the glucocorticoid receptor and identified that this interaction altered master regulators of endothelial and osteoblast differentiation. Transcriptomic information might validate our results. However, using big transcriptomic data to explore more networks or pathways is not within the scope of this study.
Reviewer 2 Report
The manuscript is well-written, the data are interesting, and deals with Glucocorticoid-induced bone loss, a severe and toxic effect of long-term glucocorticoid therapy.
Mayor concern:
par 2.1 The mice's gender is not reported, please clarify
par 2.10 The statistical analysis related to the Invivo experiments is poorly described. The power of this study is not reported, please clarify and calculate it.
Author Response
Reviewer 2
The manuscript is well-written, the data are interesting, and deals with Glucocorticoid-induced bone loss, a severe and toxic effect of long-term glucocorticoid therapy.
Mayor concern:
1.“par 2.1 The mice's gender is not reported, please clarify”
We added “Since glucocorticoid-induced osteoporosis increases the incidence of fractures in both men and women [1,2], mixed genders of mice were used in each experimental group.” To the 2.1 section.
2.“par 2.10 The statistical analysis related to the In vivo experiments is poorly described. The power of this study is not reported, please clarify and calculate it.”
We added “We used pwr R package to calculate the size of the effect and the statistical power as previously published [23]. The number of animals in each group was sufficient to reach more than 80% power to identify differences in the results.” To the 2.10 section.
Reviewer 3 Report
Comments:
1. Could the authors please clarify how the numbers of animals for each of their experiments were determine i.e. how was their study powered?
2. How many animals were used in total to complete their study?
3. Could the authors please justify the use of parametric statistics for the data anlyses?
As above.
Author Response
Reviewer 3
- “Could the authors please clarify how the numbers of animals for each of their experiments were determine i.e. how was their study powered?”
We added “We used pwr R package to calculate the size of the effect and statistical power as previously published [23]. The number of animals in each group was sufficient to reach more than 80% power to identify differences in the results.” To the 2.10 section.
- “How many animals were used in total to complete their study?”
We added “The n represented the number of animals that were used in each experimental group.” To the 2.10 section.
- “Could the authors please justify the use of parametric statistics for the data anlyses???”
We performed normality test using the Shapiro-Wilk’s method, the data in each group was normally distributed.
Round 2
Reviewer 2 Report
Despite some calculations of the Power that has been added to the manuscript, still the in vivo data are presented as aggregated mixing of male and female data.
I strongly suggest reorganizing the manuscript to describe the female and male data separately
Author Response
Reviewer 2
“Despite some calculations of the Power that has been added to the manuscript, still the in vivo data are presented as aggregated mixing of male and female data. I strongly suggest reorganizing the manuscript to describe the female and male data separately.”
This manuscript is a related paper of our recent publication (Cell Transitions Contribute to Glucocorticoid-Induced Bone Loss. Qiao X, et al. Cells. 2023 Jul 8;12(14):1810.), in which we used mixed genders of mice in each experimental group to demonstrate the osteoblast-endothelial transitions in glucocorticoid-induced bone loss. In the current manuscript, we used similar experimental design and data analysis to validate this scientific discover and identify the role of aurora A in this unwanted cell transition. The effect size and statistical power of the animal study were calculated as in the related paper. To distinguish if the gender affects the osteoblast-endothelial transitions is not within the scope of this study.
Reviewer 3 Report
The authors have satisfactorily responded to my comments.
As above.
Author Response
Thank you!